# Selective COVID-19 Coinfections in Diabetic Patients with Concomitant Cardiovascular Comorbidities Are Associated with Increased Mortality

**DOI:** 10.3390/pathogens11050508

**Published:** 2022-04-25

**Authors:** Kamaleldin B. Said, Ahmed Alsolami, Fawwaz Alshammari, Fayez Saud Alreshidi, Anas Fathuldeen, Fawaz Alrashid, Abdelhafiz I. Bashir, Sara Osman, Rana Aboras, Abdulrahman Alshammari, Turki Alshammari, Sultan F. Alharbi

**Affiliations:** 1Department of Pathology and Microbiology, College of Medicine, University of Ha’il, Ha’il 55476, Saudi Arabia; s201805660@uoh.edu.sa (A.A.); s201803781@uoh.edu.sa (T.A.); S20200579@uoh.edu.sa (S.F.A.); 2Department of Genomics, Bioinformatics and Systems Biology, Carleton University, 1125 Colonel-By Drive, Ottawa, ON K1S 5B6, Canada; 3ASC Molecular Bacteriology, McGill University, 21111 Lakeshore Rd, Montreal, QC H9X 3L9, Canada; 4Department of Internal Medicine, College of Medicine, University of Ha’il, Ha’il 55476, Saudi Arabia; a.alsolami@uoh.edu.sa; 5Department of Dermatology, College of Medicine, University of Ha’il, Ha’il 55476, Saudi Arabia; fawwazf@liveuohedu.onmicrosoft.com; 6Department of Family, Community Medicine, College of Medicine, University of Ha’il, Ha’il 55476, Saudi Arabia; fs.alreshidi@uoh.edu.sa (F.S.A.); r.alghamdi@uoh.edu.sa (R.A.); 7Department of Plastic Surgery, College of Medicine, University of Ha’il, Ha’il 55476, Saudi Arabia; a.fathuldeen@uoh.edu.sa; 8Department of Surgery, College of Medicine, University of Ha’il, Ha’il 55476, Saudi Arabia; fa.alrashid@uoh.edu.sa; 9Department of Physiology, College of Medicine, University of Ha’il, Ha’il 55476, Saudi Arabia; ah.bashir@uoh.edu.sa; 10Department of Internal Medicine, Winchester Hospital 41 Highland Avenue, Winchester, MA 01890, USA; Ssosman@dha.gov.ae

**Keywords:** COVID-19 aggravation, comorbidity, coinfections, Gram-negative coinfections

## Abstract

Coinfections and comorbidities add additional layers of difficulties into the challenges of COVID-19 patient management strategies. However, studies examining these clinical conditions are limited. We have independently investigated the significance of associations of specific bacterial species and different comorbidities in the outcome and case fatality rates among 129 hospitalized comorbid COVID-19 patients. For the first time, to best of our knowledge, we report on the predominance of *Klebsiella pneumoniae* and *Acinetobacter baumannii* in COVID-19 non-survival diabetic patients The two species were significantly associated to COVID-19 case fatality rates (*p*-value = 0.02186). Coinfection rates of *Klebsiella pneumoniae* and *Acinetobacter baumannii* in non-survivors were 93% and 73%, respectively. Based on standard definitions for antimicrobial resistance, *Klebsiella pneumoniae* and *Acinetobacter baumannii* were classified as multidrug resistant and extremely drug resistant, respectively. All patients died at ICU with similar clinical characterisitics. Of the 28 major coinfections, 24 (85.7%) were in non-survivor diabetic patients, implying aggravating and worsening the course of COVID-19. The rates of other comorbidities varied: asthma (47%), hypertension (79.4%), ischemic heart disease (71%), chronic kidney disease (35%), and chronic liver disease (32%); however, the rates were higher in *K. pneumoniae* and were all concomitantly associated to diabetes. Other bacterial species and comorbidities did not have significant correlation to the outcomes. These findings have highly significant clinical implications in the treatment strategies of COVID-19 patients. Future vertical genomic studies would reveal more insights into the molecular and immunological mechanisms of these frequent bacterial species. Future large cohort multicenter studies would reveal more insights into the mechanisms of infection in COVID-19.

## 1. Introduction

Several factors are known to influence the continuous outbreak of the SARS-CoV2 pandemic. These include the speed, the surging waves of re-emergences, and lack of specific widely available treatment, which all aggravated the devastating COVID-19 coronavirus pandemic. For instance, the enhanced epidemicity has resulted in a total of 312,173,462 confirmed cases of COVID-19 globally, including 5,501,000 deaths reported to the World Health Organization (WHO) as of 12 January 2022 (available at https://COVID-19.who.int/; accessed on 12 January 2022). The increased human–human transmission has caused the re-emergence of variants and sub-clonal populations in different regions despite the clonal genome [1,2]. However, an unprecedented global effort led to efficient vaccine products such as Pfizer Biontech, Moderna, and AstraZeneca. This was followed by the biggest mass vaccination campaigns ever witnessed, resulting in a total of 9,194,549,698 vaccine doses administered. Despite the high efficacy and safety of the vaccine, many side effects were reported [3,4,5,6,7]. Thus, in the absence of widely available medicine, significant gaps remain in our understanding of the mechanisms of SARS-CoV2 virulence, secondary coinfections, and patient comorbidities that all significantly influenced disease outcomes. 

Despite the role of coinfections as important contributors to mortality in viral pandemics, they are not well understood in SARS-CoV2 outbreaks. Recent findings revealed that other respiratory viral coinfections are likely to infer cross-protections [8,9,10]. Furthermore, common viral coinfections negatively correlated in COVID-19 disease implying potential cross-protection and highlighting the significance of screening for bacterial respiratory coinfections instead [10]. Therefore, bacterial coinfections and superinfections are potential aggravators; however, no clear consensus on empirical antimicrobial therapy is available. For instance, rapid review meta-analysis of 3338 COVID-19 patients identified that 14.3% had bacterial coinfections; however, most patients received empirical therapy that they did not need [11]. In addition, significant progress was made in understanding coinfections in previous viral pandemics such as SARS-1 and Middle East respiratory syndrome (MERS). Low levels of bacterial and fungal coinfections were reported early in the COVID-19 pandemic, and most of them were in China (2020) [12]. Cases of coinfections with single bacterial species such as *Staphylococcus aureus* have also been reported in comorbid patients of COVID-19 [13]. A few experimental studies demonstrated that the coinfection and competition between pathogens inside the host tissues can have major consequences for their transmission success [14]. Perhaps one of the most catastrophic coinfections was the emergence of mucormycosis during the second wave of the COVID-19 pandemic [15]. Thus, coinfections are important factors in developing strategies for antibiotic empirical therapy and patient prognosis during COVID-19. Although some strategies such as empiric monotherapy were suggested, many authors recommended confirmation by larger studies to assess the actual prevalence and the predictors of co-infection together with its prognostic impact on critically ill patients [16,17]. However, the frequency and types of coinfections would potentially differ in different patients’ clinical profiles and comorbidities. 

Diabetic patients are widely reported in many countries, including Saudi Arabia as a higher risk for mortality than their non-diabetic counterparts for COVID-19 [18,19,20,21,22]. However, most of the studies had limitations. For example, some were single-center, others had small samples sizes, and still others measured on single factors. In addition, the data on the correlation of diabetes to the poor outcomes for COVID-19 are quite conflicting. A recent meta-analysis exploring the relationship between diabetes and COVID-19 mortality and severity reported the association with a two-fold increase in mortality, as well as severity of COVID-19, as compared to non-diabetics [23]. However, a letter to the editor identified several limitations to the aforementioned article [24]. Similarly, in a meta-analysis of hospitalized patients in China, the COVID-19 mortality rate was 9.9%. In addition, a higher prevalence of diabetes mellitus was independently associated with a worse prognosis. The authors argued that this independent influence of diabetes mellitus should be viewed as hypothesis and suggested further studies [25]. Discussion on the correlations of diabetes to COVID-19 prognosis without the context of bacterial coinfection is incomplete. The bacterial coinfections and their empirical antibiotic therapies are significant components of the mechanisms of diabetes that largely determine severity and outcome. Although diabetes strongly influences COVID-19 outcomes, the mechanisms underlying increased susceptibility to infection have not been well defined beyond the known glucose metabolism. Even though it has been long established that lymphocytes upregulate insulin receptors upon immune stimulation, dysregulated insulin signaling in immune cells is the potential mechanism associated with diabetes infections [26]. As such, the impaired innate and adaptive immune responses and the low-grade inflammatory response may protect the microbes by the deficiency of microRNA-146a leading to sever outcomes [27,28]. Thus, to prove these mechanisms, significant correlations are needed on the influence of diabetes and coinfections during COVID-19 infections. Nevertheless, the complicating factor in COVID-19 is that coinfection rates are much lower contrary to that in other influenza pandemics. In addition, studies describing the types, resistant profiles, and extent of different bacterial species implicated in COVID-19 patient outcome is limited.

Coinfection rates in hospitalized hypertensive patients is not widely reported. However, recent study reported 75% of the overall coinfection in hypertensive patients out of 144 cases (*p*-value = 0.003), where almost half (48 cases) originated in the community (*p* = value 0.03) [29]. The COVID-19 impact on the kidney was often in the form of acute kidney injury (AKI), which is an independent risk factor for mortality. Although it has not been widely studied, the AKI is potentially due to direct cytotropic effect and cytokine-induced systemic inflammatory response resulting in continuous renal replacement therapy as the common practice followed [30]. Acute kidney injury showed 5–15% of the cases during SARS-CoV and MERS-CoV and a higher mortality rate of 60 to 90% [31]. The exact mechanism of kidney injury was not fully understood initially [32]; however, sepsis leading to cytokine storm syndrome or direct cellular injury due to the virus has been reported. Nevertheless, while studies on AKI are on the rise, COVID-19’s impact on patients with underlying chronic kidney disorder is not fully understood [30].

COVID-19 potential risk factors included male gender and comorbidities such as hypertension, heart disease, diabetes, and malignancy [33,34]. There are not much data available on COVID-19 and chronic liver diseases. The tropism of the virus in liver cells is also not well studied. Nevertheless, the tissue predilection sites for SARS-CoV-2 replication have not been fully elucidated for difficulties in obtaining biopsy samples and the requirement for high level laboratory containment facilities. Attempts were made to understand the virus mechanisms of liver pathogenicity. Since the spike protein binds ACE2 to gain cell entry where transmembrane serine protease 2 (TMPRSS2) and paired basic amino acid cleaving enzyme (FURIN) are important for infection, expression levels of the latter two were studied. The TMPRSS2 and FURIN showed a broad gene expression profile across many liver cell types [35]. However, in a combined analysis of three single-cell RNA sequencing datasets from liver tissue of healthy individuals, very few hepatocytes co-expressed ACE2 and TMPRSS2 [36]. Since there was variation in expression levels across different liver models used, more insights are needed for SARS-CoV-2 hepatotropism due to the poor outcomes and high rates of hepatic decompensation in patients. 

The role of SARS-CoV-2 virus in inducing stroke has been widely proposed; however, the mechanisms involved is not fully understood. The nasal olfactory bulb expresses varying transcript levels for nasal partitioning ration-inspiration (NRP1), ACE2, CD147, TMPRSS2, and Furin, which may explain the disturbance in smell and taste [37,38,39]. High expression of NRP1 in the SARS-CoV-20-infected cells of the olfactory epithelium has been found [39], indicating a potential route to complications causing stroke in COVID-19 patients. Hematogenous route for neuro-invasion involves the infection of mucosal linings, giving access to the lymphatic system and the bloodstream where endothelial cells express the SARS-CoV-2 receptors [40,41]. These pathways may disseminate the virus to the peripheral tissues and cardiovascular systems leading to stroke.

Thus, for more insight into COVID-19 disease patterns, pathogenicity, and patients’ outcomes, detailed mechanisms of co-bacterial infection in comorbidities are urgently needed. This is because SARS-CoV2 is potentially becoming endemic in the world likely to remain as a significant health crisis if not dealt with in deep detail. Cox et al. (2020) described the incidence, prevalence, and characteristics of bacterial infection in COVID-19 patients as potentially lethal and an important unexplored knowledge gap [42]. Thus, it is of paramount importance to gain deep insight into the factors responsible for the aggravation and progression of the disease including correlations of coinfections and comorbidities. The aim of this study was to investigate the role of coinfections in the aggravation of COVID-19 in comorbid patients’ outcomes.

## 2. Results

### 2.1. Influence of Coinfections and Comorbidities in Overall COVID-19 Case Fatality Rates 

To understand the overall influence of coinfections on non-comorbid patient outcome, we examined 300 patients for COVID-19 case fatality rates (CFR). We found that CFR for bacterial co-infections in COVID-19 was (31.2%) highly statistically significant in comparison to that in non-coinfected patients (9.9%). This implied that bacterial infections are importantly associated with the COVID-19 outcome. The overall coinfection rate was (36%, *n* = 109) among 301 COVID-19 patients. However, to understand the influence of 7 different comorbidities in CFR, we examined 129 comorbid and coinfected COVID-19 patients. No steroids or other predisposing drugs for infections were prescribed for these COVID-19 patients. The detailed statistical analysis carried out is given in the Additional File and Statistics File. As shown in Table 1, 85.7% (*n* = 24 out of 28) of COVID-19 non-survivor senior patients mainly had diabetes and coinfections with different species, namely, MDR *K. pneumoniae* (93%) and XDR *A. baumannii* (73%). In *Klebsiella pneumoniae* co-infected diabetic patients, the most frequent heart comorbidities were asthma (50%), hypertension (93%), and ischemic heart disease (86%). Similarly, in *A. baumannii* co-infected diabetic patients, the most frequent other heart comorbidities were asthma (45.4%), hypertension (64%), and ischemic heart disease (54.4%). Appendix A shows details of the combinations of comorbidities associated to diabetic patients in non-survivors who were coinfected with different pathogens. 

The resistance classification of the species in this study was based on the standard definitions of resistances in different antimicrobial categories used as indicated in the Material and Methods section. The antimicrobial susceptibility testing for the Gram-negative bacteria was performed for 21 drugs in different categories, as shown in Material and Methods. *Klebsiella pneumoniae* was the most frequent COVID-19 coinfecting pathogen. Over half of its isolates were highly resistant to many drugs, including AUG, amoxicillin*/clavulanic acid (2/1); ATM, aztreonam; FOX, cefoxitin; CRO ceftriaxone; AMC, ampicillin*/sulbactam; CXM, cefuroxime; KF, cephalothin; and NIT, nitrofurantoin. On the other hand, high susceptibility of isolates was obtained (>83%) for some antimicrobials including TGC, tigecycline; CS, colistin; and AK, amikacin. Intermediate resistances were also seen for NIT, nitrofurantoin; TZP, tazobactam; IMI, imipenem; CIP, ciprofloxacin; and AUG, amoxicillin. Patients with *K. penumoniae* seemed more exhausted showing heavy productive cough unlike that of dry COVID-19. All non-survivors died at ICU and had typical COVID-19-compatible symptoms such as infiltration in CXR, shortness of breath requiring higher oxygenations, and low absolute lymphocyte counts (LALC) below (<5) (Table 1). Microbiological characteristics and disease patterns implied infection with a single strain. All laboratory cultures showed more profound colonies with heavily diffused growth appearance on agar plates within 18 to 24 hours of incubation. *Acinetobacter baumannii* ranked second in nosocomial pathogens (Table 1). According to standard resistance definitions of classifications, *A*. *baumannii* was classified as extremely drug resistant. This organism was resistant to nearly all antibiotics tested for treatment, exhausting all available drugs except for colistin, for which it was almost fully susceptible. Non-survivor patients with *A.baumanni* were mostly seniors who died at ICU. Clinical characteristics and oxygen support was similar to that of *K. pneumoniae.* In addition, three non-survivor patients coinfected with MDR *E. coli* and the one who had XDR *P. aeruginosa*, were all diabetic (100%) (Table 1). 

### 2.2. Diabetes

We have studied 129 coinfected comorbid patients for the influence of underlying causes in COVID-19. In a COVID-19 patient population sample of 101 that were coinfected with different bacterial pathogens, 80.2% (*n* = 81) had underlying diabetes and the rest of the patients (*n* = 28) did not quality for inclusion criteria. In the diabetic population, the case fatality rate (CFR) was 35.8% compared to 10 % in the non-diabetes population (Figure 1a). We found significant association between diabetes and CFR in COVID-19 patients who had underlying bacterial coinfections (*p*-value = 0.052). In this study, 90% of patients without diabetes survived; however, 64.2% diabetic patients did not survive. Association studies to establish relationships between CFR and any specific bacteria species was not significant (*p*-value = 0.052). Albeit *K. pneumoniae* and *A. baumannii* each caused an equal number of deaths (50% each species) (Figure 1b,c).

### 2.3. Hypertension (HTN)

In this study, the CFR in hypertensive COVID-19 patients who had bacterial coinfection was 50%, while in those with no hypertension, the CFR was 5% (Figure 2a). We found this association significant, as indicated by the *p*-value (*p*-value = 2.49961772559523 × 10^−6^), in a COVID-19 patient population of 98, where 59.1% (*n* = 58) were hypertensive. In other words, 95% of non-hypertensive patients survived COVID-19. In the analysis of recovery and death rates among COVID-19 patients with hypertension and bacterial coinfection, the association of CFR to specific bacterial species was not significant (*p*-value = 0.227). However, the death rates were higher in *K. pneumoniae* (60.9%) and *A. baumannii* (57%) than other bacterial coinfections (Figure 2a,b). A total of 31 patients in the population of 129 did not qualify for inclusion in this comorbidity.

### 2.4. Ischemic Heart Disease (IHD)

In the 101 coinfected COVID-19 patients studied, 39% had underlying ischemic heart disease. The rest of the patients did not satisfy some of the inclusion criteria. The association of CFR of ischemic heart diseases in these coinfected COVID-19 patients was significant (*p*-value = 7.75228536995411 × 10^−9^). The CFR was 64% in coinfected patients with ischemic heart and 9.7% in those without the disease. Over 90% of patients without IHD survived COVID-19 infection. In addition, the analysis of association of death and recovery among different patients coinfected with different bacterial species showed significant association between CFR and higher frequencies of *K. pneumoniae* and *A. baumannii* (*p*-value = 0.02186) compared to other species examined in this study. In this group of patients (101), we also studied the influence of heart failure in the COVID-19 CFR; however, there was no significant association due to the small sample size 3% (*n* = 2.97) (Figure 3a,b).

### 2.5. Bronchial Asthma

Out of 97 COVID-19 patients (the rest of the patients did not satisfy major inclusion criteria) coinfected with different bacterial pathogens, 39.2% (*n* = 38) were asthmatic. The CFR among the patients with asthma was 44.7% compared to 23.7% of those in the same group but without asthma comorbidity (Figure 4a). The Association of the COVID-19 CFR in patients with asthma was significant, as indicated by the *p*-value = 0.03. Furthermore, the survival rates in those patients with no asthma was 76.3%, while in asthma patients, it was 55.3% (Figure 4a). The association of COVID-19 CFR to specific coinfecting bacterial species was not significant (*p*-value = 0.175), indicating that any bacterial coinfection studied was equally significant. However, the higher frequency of death was shown in patients with higher *K. pneumoniae* (63.6%) and *A. baumannii* (54.5%) coinfections. Patients who did not pass important inclusion criteria were removed from the study. 

### 2.6. Chronic Kidney Disease (CKD)

In the population of 101 coinfected COVID-19 patients, 17% (*n* = 17) had underlying CKD. The analysis of these patients showed that the COVID-19 CFR was 76.5% in those with CKD and coinfections compared to 21.4% for those with no CKD. The association was significant as indicated by the *p*-value = 7.21169363455566 × 10^−6^. However, the association was not significant in relation to specific bacterial species, as indicated by the nonsignificant *p*-value = 0.643. The death rate was 76.5% in CKD patients, which was not associated to any specific bacterial species as indicated by the non-significant *p*-value = 0.643 (Figure 5a,b).

### 2.7. Chronic Liver Disease CLD

The number of CLD patients in a population of 101 bacterial coinfected COVID-19 patients analyzed was 13% (*n* = 13). The estimation of CFR was 92% among those with CLD, while it was 21% among those with no CLD. This association was significant (*p*-value = 2.46689377316463 × 10^−7^). However, the small sample size (*n* = 13) made the association not significantly related to any specific bacterial species (*p*-value = 0.202) (Figure 6a,b).

## 3. Discussion

In this study, we have investigated the associations and case fatality rates (CFR) in bacterial coinfected COVID-19 hospitalized patients with underlying comorbidities. To first understand the influence of bacterial coinfection per se in the course of this viral infection, we have determined the COVID-19 CFR in coinfected non-comorbid patients using a control group of non-coinfected patients. Using 300 patients, we first established that the influence of coinfection in the CFR was highly significant (z-value = 3.1); the overall rate was 36% (*n* = 109 of 300), while the CFR was 31.2% in coinfections and 9.9% in non-coinfected patients. This work examined the associations of specific bacterial species in the CFR of comorbid patients. The analysis of comorbidities in hospitalized patients and CFR association to specific bacterial species revealed more insights into the disease profiles and pathogenicity of the virus. In addition, we identified diabetes as the most frequent (85% Table 1) and the highest associated comorbidity to different types of bacterial coinfection in those who did not survive. For this reason, we used diabetes against which to understand its association to coexisting heart comorbidities, bacterial coinfections, and patients’ outcomes.

The high CFR among coinfected diabetic patients (35.80%) compared to that in nondiabetics (10%) was not associated to any particular bacterial species (*p*-value = 0.052 not significant) indicating the influence of common coinfection in the outcome. Although many reported on the correlation of diabetes to mortality [18,19,20,21,22] studies on the COVID-19 coinfection of diabetic patients are limited globally. Specifically, despite the high rates of endemic diabetes in the region, data are limited on the coinfections of COVID-19 patients with underlying diabetes. The mechanisms of coinfection in COVID-19 and the corresponding guidelines for the use of empirical therapy are not yet clear and vary in different countries. In Singapore for instance, 19 of 240 (7.9%) hospitalized patients received bacterial antibiotics [43]. Recent theoretical studies revealed elevated blood glucose as a key facilitator in the progression of COVID-19 by enabling increased immune evasion, aggressive invasion, and cytokine storm [44]. It is plausible that bacterial superinfections would be more advantageous under these conditions. 

In this study, the increased rates of coinfections by *K. pneumoniae* and *A. baumannii* in diabetics and the highly significant association with COVID-19 CFR is consistent with aggravations in the aforementioned altered glucose metabolism and oxidative stress. Reports on selective coinfections with single bacterial species, such as *Staphylococcus aureus*, have also been reported in comorbid patients of COVID-19 [13]. It has been shown experimentally that coinfection and competition between pathogens inside the host tissues can have major consequences for their transmission success [14]. Perhaps one of the best examples was the emergence of mucormycosis during the second wave of the COVID-19 pandemic [15]. Since diabetes is a significant contributor to mortality [45] in the region, it is important to understand the mechanisms of COVID-19 coinfections in comorbid patients. Interestingly, among all comorbidities in the current study, heart-related problems, including ischemic heart disease, hypertension, and asthma, were more frequent in diabetic patient consistent with earlier finding [46]. Similarly, an 11-year retrospective study on 243 diabetic foot ulcer patients at King’s College Hospital indicated that ischemic heart disease was the major cause of death in 62.5% of the studied patients [47]. However, the significant role of coinfection in worsening COVID-19 outcome in hospitalized comorbid patients has been identified in this study. Although Saudi Arabia is ranked seventh in type 2 diabetes among the top 10 countries, with the prevalence rate of 25.4% [48], the prevalence of diabetic foot remains similar to global rates [49]. 

Significant association of COVID-19 CFR to specific coinfections of *K. pneumoniae* and *A*. *baumannii* among patients with ischemic heart disease was found in this study (*p*-value = 0.02186). Although ischemic heart disease CFR (64%) was significantly associated (*p*-value = 7.75228536995411 × 10^−9^), the role of the wo species in the fatal outcomes is profoundly consistent with their aggressive coinfection and clinical pattern. *K. pneumoniae* is an extremely resilient and defensive bacterium that appears to elusively deregulate the immune system and evade to survive. Many bacterial and host factors have been independently identified for these mechanisms [50]; however, less is known about *K. pneumoniae* associations in COVID-19 patients with underlying ischemic heart disease. Since intubation and lethal oxygen interventions are the last resorts in the fight against the already established infection, aspiration pneumoniae is not likely to be the primary cause for the selective growth of *K. pneumoniae* and *A*. *baumannii* [51]. Coinfections of the latter species as well as *P. aeruginosa*, the poor carbohydrate fermenters, in diabetes may support the alternative mechanisms than the sugar metabolism *per se*. *Acinetobacter baumannii*, although not the most frequent pathogen in this study, is known for its highest mortality rates in nosocomial infections in agreement with earlier findings [52,53]. In addition, while this organism has been reported in many countries as coinfecting COVID-19 patients including Wuhan (China), France, Spain, Iran, Egypt, New York (USA), Italy, and Brazil [54], it is not mentioned as a primary species in contrast to our study. However, a retrospective COVID-19 study in in Spain showed multidrug-resistant *A. baumannii* as the leading agent in respiratory infections and bacteremia with outbreak [55].

Thus, in this study, we present findings related to frequent bacterial coinfections and underlying comorbidities and their association to the disease outcome in hospitalized COVID-19 patients. For the first time, to the best of our knowledge, we report on the predominance of limited Gram-negative bacterial COVID-19 coinfections in non-survival diabetics who had concomitance of heart-related comorbidities. *Klebsiella pneumoniae* and *A. baumannii* were the most frequent species associated to high mortality rates. Other bacterial species and comorbidities did not have significant correlation to the patients’ outcomes. This has significant clinical implications in the treatment strategies of COVID-19 patients. This study is limited by confinement to single center study; a large cohort surveillance of COVID-19 bacterial coinfections from comorbid patients in different hospitals would gain more insights. Strict inclusion criteria excluded may patients who were otherwise might have been considered beginning comorbid disorder. Future vertical studies on large cohort hospitals would reveal more insights into the molecular and immunological mechanisms of these frequent bacterial species. 

## 4. Materials and Methods

### 4.1. Study Descriptions, Hospital, Data Acquisitions, and Statistical Analysis

This work was a retrospective study during the last quarter of 2021. A panel of experts screened the data for eligibility. Eligible patients were those with confirmed positive PCR for SARS-CoV-2 as the major inclusion criterion. This implied testing positive for COVID-19 confirmed by clinically compatible symptoms (and epidemiological criteria for non-symptomatic carriers) and substantiated by standard molecular tests using the approved real-time reverse transcription PCR (RT-PCR) performed on nasopharyngeal throat swab specimens at the Ha’il Health Regional Laboratory (HHRL) for COVID-19. All comorbid patients included those hospitalized groups for more than 48 to 72 h, must have been isolated in KSSH center, and must have confirmed clinical data on comorbidity disorders. Unclear or uncharacterized test results were removed. The basic demographics and clinical characteristics (underlying diseases, comorbidities, invasive procedures, outcome status, and survival characteristics) of patients coinfected or have developed underlying comorbidities were collected. In addition, patients were ineligible if they had developed any non-COVID, non-specified comorbidity other than the major underlying comorbidities identified and for which they were hospitalized, namely, diabetes, asthma, ischemic heart conditions, hypertension, heart failure, and chronic kidney and liver diseases. They were also non-eligible with known community-associated infections less than 48–72 h of their admission. 

The panel reviewed the data from admission until death or discharge, including COVID-19 data. To avoid inclusion or exclusion biases, data were collected at the same time for all groups. The data sources included, but were not limited to, records of general clinical signs and major treatment data that would potentially change the course of infection (especially steroids), ecotype isolate resistance profiles in case of SARS-CoV2 bacterial coinfection, and general laboratory results, which were abstracted using a structured data collection template with a standardized data dictionary routinely used. Soft versions of e-records on hospitalized COVID-19 patients included in the study retrospectively. Details of the additional data referred to for direct involvement and confirmation on coinfections and comorbidities of survivors or on death records included, but were not limited to, the following: records of chest X-ray (CXR) used for pulmonary involvement and lung abnormalities related to COVID-19, non-invasive ventilation procedures, intubations and mechanical ventilations, lowest absolute lymphocyte count (LALC), and hematology records. 

Significance of bacterial coinfections per se on patient outcomes was determined from overall groups of non-comorbid patients. Since most comorbidities are risks for infections, selections were made for comorbid patients coinfected with different bacterial species. However, recruitment for different comorbidity disorders was studied independently, where specific criteria applied for inclusion or exclusion into a particular comorbidity analysis. For example, patients with non-clear results for diabetes were removed for analysis of outcomes for the disease.

### 4.2. Work Environment

The HHRL is a standard laboratory center, certified and accredited by the Saudi Central Board for Accreditation of Healthcare Institutions (CBAHI)-Code 2739. All patients were hospitalized at the King Salman Specialist Hospital (KSSH), Ha’il, Kingdom of Saudi Arabia (KSA). The KSSH hospital is a 500-bed tertiary care hospital designated to COVID-19 patients in addition to other specialized medical care services to Ha’il and all socioeconomic populations of the region. The KSSH is certified and accredited by the Saudi Central Board for Accreditation of Healthcare Institutions (CBAHI)-Ref.no. HAL/MOH/HO5/34213. 

### 4.3. Diagnostic Clinical Microbiological Data

Microbiological data were obtained from microbiology laboratory records, hospital medical records, and COVID-19 records in KSSH. Pathogens were identified by using routine standard bacteriological methods and ID and susceptibility testing using automated Systems. This included primarily BD Phoenix system (BD Biosciences, Franklin Lakes, NJ, USA), MicroScan plus (Beckman Coulter, Brea, CA, USA), and BD BACTEC system (BD Biosciences, Franklin Lakes, NJ, USA) for the identification and antimicrobial sensitivity analysis of microorganisms. Susceptibility was confirmed by culture and agar diffusions experiments. The susceptibility testing and breakpoint interpretive standards were carried out according to the recommendations of Clinical and Laboratory Standard Institute (CLSI document M100S-26) [56]. The following antimicrobials were tested; however, only the most effective drugs were prescribed: Antimicrobial susceptibility patterns of Gram-negative clinical isolates against 21 antibiotics, Abbreviations followed by full names: AK, amikacin; AUG, amoxicillin*/clavulanic acid (2/1); AMS, ampicillin*/sulbactam (2/1); ATM, aztreonam; FEP, cefepime; FOX, cefoxitin; CAZ, ceftazidime; CRO, ceftriaxone; CXM, cefuroxime; KF, cephalothin; CIP, ciprofloxacin; CS, colistin; ETP, ertapenem; CN, gentamicin; IMI, imipenem; LEV, levofloxacin; MRP, meropenem; NIT, nitrofurantoin; TZP, tazobactam; TGC, tigecycline; and SXT, trimethoprim/sulfamethoxazole. 

### 4.4. Standard Definitions of Resistances

Antibiogram data were used to classify pathogens as multidrug-resistant, extremely drug-resistant, or pan-drug-resistant (MDR, XDR, or PDR) according to the guidelines of the European Centre for Disease Control [57]. The MDR was defined as acquired non-susceptibility to at least one agent in three or more antimicrobial categories, XDR was defined as non-susceptibility to at least one agent in all but two or fewer antimicrobial categories (i.e., bacterial isolates remain susceptible to only one or two categories), and PDR was defined as non-susceptibility to all agents in all antimicrobial categories.

### 4.5. Criteria for Sampling Coinfection and/or Superinfection

COVID-19 patients were considered coinfected when a respiratory, blood, or relevant specimen was positive for a pathogenic organism following National Health Safety Network (NHSN) criteria [58]. However, we excluded cultures that were classified as possible or probable pneumonia or a contamination. Only confirmed COVID-19-compatible signs and specifically identified coinfections were eligible. Usually, approximately ~ 30% of patients have sputum production particularly old age. However, since SARS-CoV-2 pneumonic patients often lack respiratory secretions, specifically during the diagnostic dry cough stage), cultures were mostly performed in patients under invasive mechanical ventilation or on respiratory tract secretions. A patient would be considered co-infected with microbes during COVID-19 infection when at least one of the performed microbiological investigations isolated a pathogenic bacterium (whatever the bacterial count) or a virus. Readjusting and de-escalation of the recommended Surviving Sepsis Campaign guidelines on empiric antibiotic therapy and the management of critically ill adults with COVID-19 was followed once microbiology results were obtained. 

### 4.6. Statistical Analysis of the Data

All types of data, including clinical and microbiological, were normalized and analyzed using Statistical Package for Social Sciences software (IBM SPSS; Version 24 SPSS version 23.0 for Windows (SPSS, Inc., Chicago, IL, USA)). The analysis was descriptive and stratified; we present absolute numbers, proportions, and graphical distributions. We conducted exact statistical tests for proportions and show *p*-values where appropriate (a *p*-value < 0.05 was considered statistically significant). 

## Figures and Tables

**Figure 1 pathogens-11-00508-f001:**
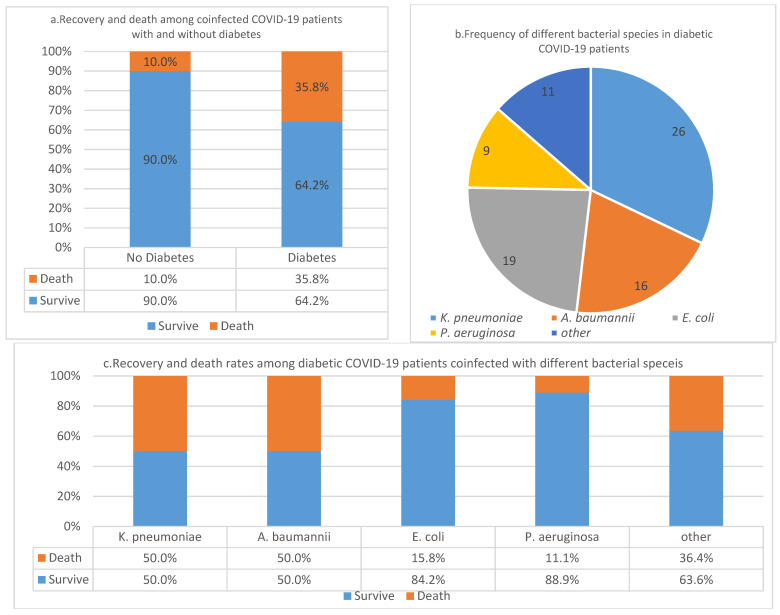
(**a**) Case fatality rates of diabetes in coinfected COVID-19 patients; (**b**) Frequency of bacterial species in diabetic COVID-19 patients; (**c**) Recovery and death rates among diabetic COVID-19 patients coinfected with different bacterial species.

**Figure 2 pathogens-11-00508-f002:**
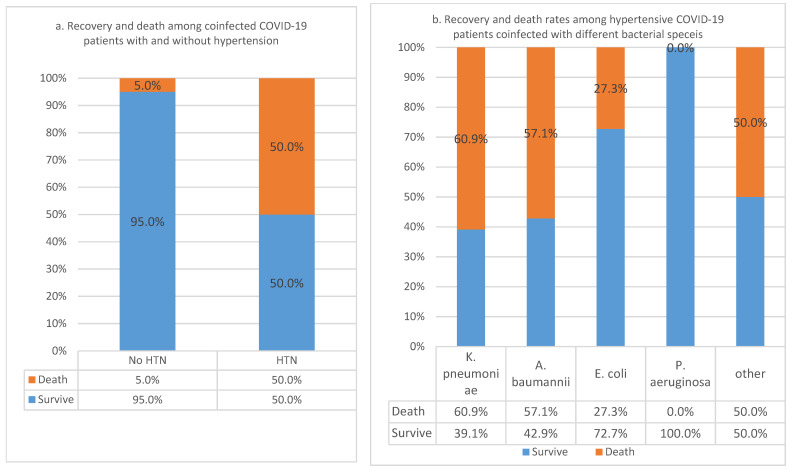
(**a**) Case fatality rates of hypertension in coinfected COVID-19 patients; (**b**) Recovery and death rates among hypertensive COVID-19 patients coinfected with different bacterial species.

**Figure 3 pathogens-11-00508-f003:**
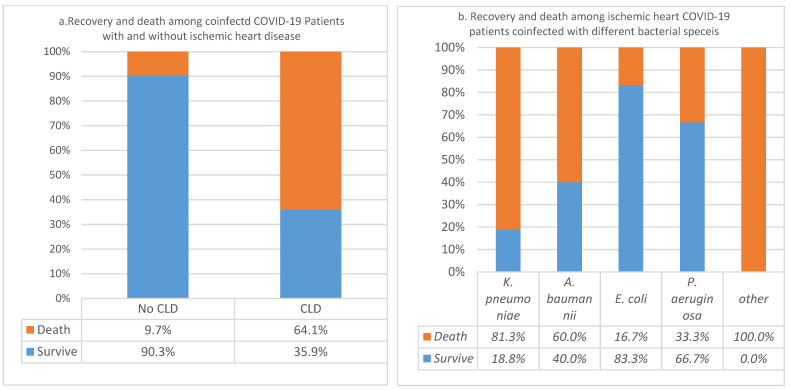
(**a**) Case fatality rates of ischemic heart disease in coinfected COVID-19 patients; (**b**) Recovery and death rates among ischemic heart COVID-19 patients coinfected with different bacterial species.

**Figure 4 pathogens-11-00508-f004:**
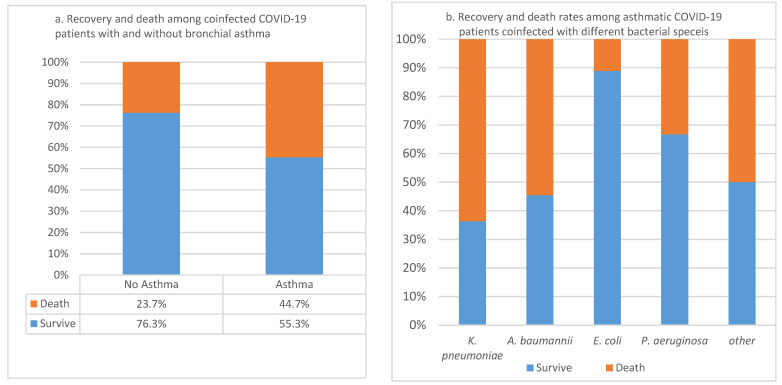
(**a**) Case fatality rates of asthma in coinfected COVID-19 patients; (**b**) Recovery and death among asthmatic COVID-19 patients coinfected with different bacterial species.

**Figure 5 pathogens-11-00508-f005:**
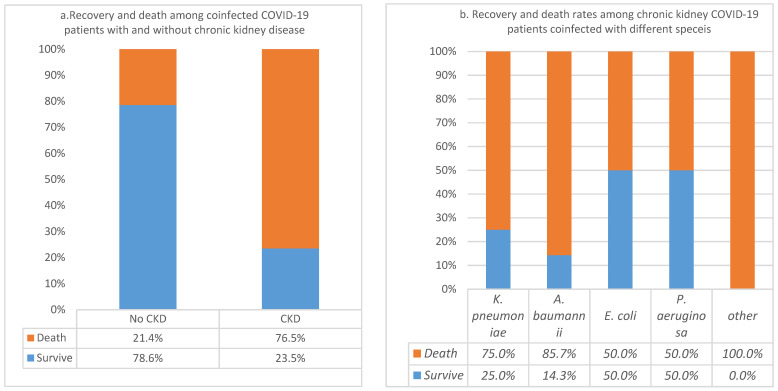
(**a**) Case fatality rates of chronic kidney diseases (CKD) in coinfected COVID-19 patients; (**b**) Recovery and death among COVID-19 patients with CKD and coinfected with different bacterial species.

**Figure 6 pathogens-11-00508-f006:**
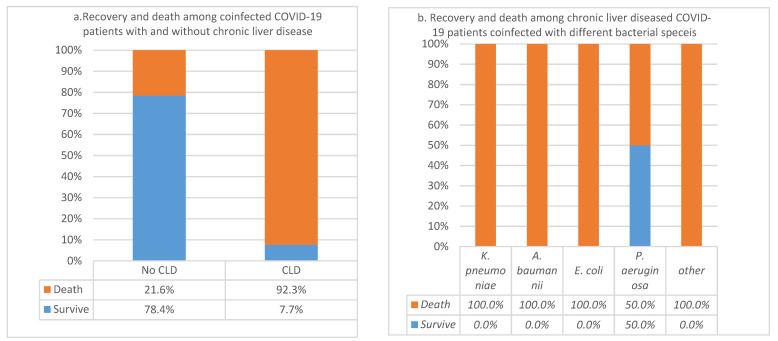
(**a**) Case fatality rates among coinfected COVID-19 patients with and without chronic liver disease; (**b**) Recovery and death among COVID-19 patients with chronic liver disease coinfected with different bacterial species.

**Table 1 pathogens-11-00508-t001:** Mortality rates and clinical characteristics of *K. pneumoniae and A. baumannii* co-infected diabetic COVID-19 patients with concomitant heart comorbidities.

Deaths in Total and Diabetic Coinfections			Concomitant Heart Comorbidities						Supportive and Clinical Characteristics						
Major coinfecting bacteria	Total Death per major coinfections	Deaths per coinfections with diabetes	Asthma	Hypertension	Chronic Kidney Diseases	Chronic Liver Diseases	Heart Failure	Ischemic Heart Disease	Patient Age Range			Infilt CXR?	Steps in High Oxygen Intervention Requirments before Death	ICU	LALC
									0–20 Year Old	21– 49 Year Old	Seniors >50 Year				
*K. pneumoniae* (MDR)	14	13 (93%)	7 (50%)	13 (93%)	3 (21%)	5 (36%)	2 (14%)	12 (86%)	0	1	13	Yes	All were initially ventilated, then required intubation and high oxygenation (>4 L on avg)	Yes	<5
*A. baumannii* (XDR)	11	8 (73%)	5 (45.5%)	7 (64%)	5 (45.5%)	2 (18%)	0	6 (54.5%	0	2	9	Yes	All, except one, required initial ventilation, then required intubation and high oxygenation (>4 L on avg)	Yes	<5
*E. coli* (MDR)	3	3	1	3	1	1	0	1	0	0	3	Yes	Only these three patients were intubated with high oxygenation before death	Yes	<5
Totals	28	24 (85.7%)	13 (46.4%)	23 (82%)	9 (32%)	8 (28.5%)	2 (7%)	19 (68%)	0	3 (10.7%)	25 (89%)				

## Data Availability

All data included in the manuscript; Appendix A are attached along with this submission.

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
