# Peer review of "Selective COVID-19 Coinfections in Diabetic Patients with Concomitant Cardiovascular Comorbidities Are Associated with Increased Mortality"

_pathogens, 2022, doi:10.3390/pathogens11050508_

Round 1

Reviewer 1 Report

Comments to the authors:

Summary: The paper evaluates the significance of associations of specific bacterial species and different comorbidities in the outcome and case fatality rates among 129 hospitalized comorbid COVID-19 patients.

The authors present findings related to frequent bacterial coinfections and underlying comorbidities and their association to the disease outcome in hospitalized COVID-19 patients. They report on predominance of limited Gram-negatives bacterial COVID-19 coinfections in non-survival diabetics who had concomitance of heart related comorbidities.

  1. Can the authors check for punctuation errors and redundant words throughout the manuscript?
  2. Please check for spelling errors (line 226 and 227) throughout the manuscript.
  3. Can the authors elaborate a bit more on the two bacterial species that were associated with high mortality rates?
  4. Can the authors comment on the impact of age and gender on the case fatality rate of coinfected comorbid COVID-19 patients and how it impacts the treatment strategies?
  5. Can the authors comment on the significance of this study with a broader design approach with large cohort hospitals from different parts of the country?

Author Response

Step by step response to reviewer comments:

Comments to the authors:

Summary: The paper evaluates the significance of associations of specific bacterial species and different comorbidities in the outcome and case fatality rates among 129 hospitalized comorbid COVID-19 patients.

The authors present findings related to frequent bacterial coinfections and underlying comorbidities and their association to the disease outcome in hospitalized COVID-19 patients. They report on predominance of limited Gram-negatives bacterial COVID-19 coinfections in non-survival diabetics who had concomitance of heart related comorbidities.

  1. Can the authors check for punctuation errors and redundant words throughout the manuscript?

Authors Response: Thank you. We checked for punctuation errors and reiterations and corrected all that we have found throughout the manuscript

  1. Please check for spelling errors (line 226 and 227) throughout the manuscript.

Authors Response: We have corrected the spelling error in these lines and throughout the manuscript

  1. Can the authors elaborate a bit more on the two bacterial species that were associated with high mortality rates?

Authors Response: thank you. We have elaborated on the two bacterial species in the manuscript

  1. Can the authors comment on the impact of age and gender on the case fatality rate of coinfected comorbid COVID-19 patients and how it impacts the treatment strategies?

Authors Response: Yes, we agree age and genders would have differences in infection rates; particularly, in gender specific organs like genital tracks and common infection patterns. However, almost all patients were respiratory with no differences in gender and almost all were at ICU, so we didn’t’ see the significance of these the infection patterns.  

  1. Can the authors comment on the significance of this study with a broader design approach with large cohort hospitals from different parts of the country?

Authors Response: this is a great question. We think this approach will be useful in first identifying the genotypes circulating and whether they belong to the pandemic hypervirulent strains being tranced globally. More important is that the design will be followed by an initiative to establish a local and regional strain profiles of these novel isolates. This would eventually follow a downstream vertical genomic analysis of dominant clones identified. These initiatives would also include the community acquired strains at a broader range for the their reginal community-transmission dynamics in the country and the region at large. These approaches would certainly shed more lights and gain significant insights into the molecular mechanisms of selection during COVID-19 co-infection, host and bacterial factors involved in the potential subtle immune evasions, as well as genome-based large scale surveillance of antimicrobial resistances that would support the concerted efforts in reducing regional pan-resistances…apologies, not sure if you requested comment in the manuscript, we could not elaborate due to text limitations...thank you!

Reviewer 2 Report

Proposed paper is quite interesting, however some revisions are needed before it can be accepted for publication:

  • In the ABS only data for diabetic patients are presented. However in the paper a lot of subclassification are presented (HT, CKD and so on). Those analysis and figure are not informative at all and not clear.
  • Furthermore, very few number of total subjects limit the interest of the paper and this regards to some sub-classification in which the number of subjects is very low. Authors should display only analysis on diabetic patients. Even more than univariate analysis showed, multivariate analysis in which comorbidities are used as covariates in the model relating the two main coinfectino with death, are needed.
  • In the model also COVID-19 infection data need to be inserted such as inflammatory markers, intubation, severity of the disease. All these variables need to be inserted in the new table 1 (see below).
  • Regarding this last point, are all the subjects from an ICU? please clarify.
  • Last column of table 1 is unusefull since it is already clear that those subjects are the non survivor. In fact, table 1 is not informative at all and should be substitute with a normale table indicating the clinical charactheristics of the patients and of their coinfection.
  • Are these bacteria resistant? in which proportion? are presence of resistance related to death?

Author Response

Proposed paper is quite interesting, however some revisions are needed before it can be accepted for publication:

Authors Response: thank you for your comment.

  • In the ABS only data for diabetic patients are presented. However in the paper a lot of subclassification are presented (HT, CKD and so on). Those analysis and figure are not informative at all and not clear.

Authors Response: Thank you. We reduced elaborations on the subclassification and removed the pie chart figure from all comorbidities to focus on main comorbidities that have significance in the patience outcome and/or coinfection. We left E. coli in the table for a comparative view of how those two species are far more aggressive than E. coli even though it is a common pathogen.

  • Furthermore, very few number of total subjects limit the interest of the paper and this regards to some sub-classification in which the number of subjects is very low. Authors should display only analysis on diabetic patients. Even more than univariate analysis showed, multivariate analysis in which comorbidities are used as covariates in the model relating the two main coinfection with death, are needed.

Authors Response: we have mainly focused on displaying diabetic patients data in detail and reduced a figure (pie charts were removed from all comorbidities) from other comorbidities in the manuscript and we left only those other data that might seem significant in explaining the specific outcomes and/or coinfections by specific pathogens. Removing figures that show low number of patients is a good idea; however, we feel that multivariate may not clearly explain the complex interplay of independent factors in the outcome such as completely different bacterial species with unique pathogenicity and virulence coinfecting comorbid patients in a novel viral disease that is not well understood. We think the study will benefit from certain degree of independent examination of each case in lieu of a complex multivariance where we do not yet understand the mechanisms of the virus-host-bacterial interactions early at this point of time in the pandemic. Multivariate analysis would be a great idea when we know a bit more and close the gaps in the host-multi-pathogen interactions all of which would certainly aggravate the disease.

  • In the model also COVID-19 infection data need to be inserted such as inflammatory markers, intubation, severity of the disease. All these variables need to be inserted in the new table 1 (see below). Regarding this last point, are all the subjects from an ICU? please clarify.

Authors Response: Thank you. We have modified Table 1 by removing the last column and the low counts in co-infections and patients numbers and adding clinical and supportive therapies. Accordingly, we have modified Table 1 title as well. We left E. coli in the table for a comparative view of how those two species are far more aggressive than E. coli even though it is a common pathogen.

  • Last column of table 1 is unusefull since it is already clear that those subjects are the non survivor. In fact, table 1 is not informative at all and should be substitute with a normale table indicating the clinical charactheristics of the patients and of their coinfection.
  • Are these bacteria resistant? in which proportion? are presence of resistance related to death?

Authors Response: we have modified Table 1 significantly to make it more informative and clear by adding clinical characteristics and supportive data and removing some the items that seemed not clear. We have also added resistance and patterns of antimicrobials in the text.

Please be advised that in line 179 that was read as “As shown in Table 1, overall 85% (n = 29) of Covid-19 non-survivor senior patients mainly had diabetes and coinfections”  but will change to (85.7%, n=24 diabetics /28 total instead of 34) since we removed other bacterial spp because they were too low to include as recommended by the reviewer. 

Round 2

Reviewer 2 Report

Authors replies to all the query raised and paper improves and can now be accepted for pubblication.